# Anxiety Linked to COVID-19: A Systematic Review Comparing Anxiety Rates in Different Populations

**DOI:** 10.3390/ijerph19042189

**Published:** 2022-02-15

**Authors:** Hafsah Saeed, Ardalan Eslami, Najah T. Nassif, Ann M. Simpson, Sara Lal

**Affiliations:** 1Neuroscience Research Unit, School of Life Sciences, University of Technology Sydney, Sydney, NSW 2007, Australia; Hafsahsaeed@outlook.com.au (H.S.); Ardalan.Eslami@uts.edu.au (A.E.); 2School of Life Sciences, University of Technology Sydney, Sydney, NSW 2007, Australia; Najah.Nassif@uts.edu.au (N.T.N.); Ann.Simpson@uts.edu.au (A.M.S.)

**Keywords:** COVID-19, anxiety, mental health, qualitative systematic review

## Abstract

The COVID-19 pandemic has incited a rise in anxiety, with uncertainty regarding the specific impacts and risk factors across multiple populations. A qualitative systematic review was conducted to investigate the prevalence and associations of anxiety in different sample populations in relation to the COVID-19 pandemic. Four databases were utilised in the search (Medline, EMBASE, CINAHL, and PsycINFO). The review period commenced in April 2021 and was finalised on 5 July 2021. A total of 3537 studies were identified of which 87 were included in the review (sample size: 755,180). Healthcare workers had the highest prevalence of anxiety (36%), followed by university students (34.7%), the general population (34%), teachers (27.2%), parents (23.3%), pregnant women (19.5%), and police (8.79%). Risk factors such as being female, having pre-existing mental conditions, lower socioeconomic status, increased exposure to infection, and being younger all contributed to worsened anxiety. The review included studies published before July 2021; due to the ongoing nature of the COVID-19 pandemic, this may have excluded relevant papers. Restriction to only English papers and a sample size > 1000 may have also limited the range of papers included. These findings identify groups who are most vulnerable to developing anxiety in a pandemic and what specific risk factors are most common across multiple populations.

## 1. Introduction

Infectious disease outbreaks have plagued human history for millennia, with an occurrence not unknown to man, the effects of these outbreaks have eluded many. With the complexities of society, there are a plethora of ways these events may cause mental turmoil. As defined by the Diagnostic and Statistical Manual of Mental Disorders (DSM-5), anxiety is a state of excessive fear that translates to behavioural disturbances [1]. Anxiety has been linked to increased ulcers, back issues, migraines, and asthma [2]. In extreme cases, it is an independent risk factor for heart disease [3]. Perpetuated by stressful environments, anxiety threatens wellbeing when worry and fear regarding real or perceived threats hijacks an individual’s ability to regulate these emotions. Infectious disease outbreaks often evolve into epidemics or pandemics, which bring about financial instability, quarantine and lockdowns, social isolation, and complete disturbance of the norm. It is in this state of pandemonium that mental health deterioration may occur.

Officially declared by the World Health Organisation (WHO) as a pandemic in March 2020 [4], COVID-19 has transformed the way the world functions and triggered an altered perception of the effects and consequences of infectious disease. Originating in Wuhan, China, COVID-19 has spread rapidly worldwide, with 4,574,089 globally reported deaths as of September 2021 [4]. An epidemiological measurement called the basic reproduction number, or R0, is the average number of secondary cases that are derived from a single primary infection, with any number over one causing exponential infection growth [5]. With an average R0 of 3.38, COVID-19 is highly transmissible [6]. This transmissibility has resulted in astonishing rates of infection and has placed a massive demand on hospital resources, challenging even the most established healthcare systems [7]. The physical manifestations of COVID-19 are apparent in the overburdened hospitals and long-lasting adverse effects of the disease. The scale of infection has been linked to psychological distress, implying something sinister may be emerging, a mental health crisis [8].

Past infectious disease outbreaks, such as the severe acute respiratory syndrome (SARS), swine flu (H1N1), and Ebola, have, in each case, demonstrated an increased prevalence of anxiety [9,10]. In the last two years, similar findings have been widely published regarding the COVID-19 pandemic [11]. A delineation between the COVID-19 pandemic and past infectious disease outbreaks are apparent through the unprecedented implementation of lockdowns, social isolation, and quarantines effecting the global populace. The Australian Bureau of Statistics (ABS) reported that the incidence of anxiety had doubled in 2020 compared to previous years [12]. A longitudinal study conducted in the United Kingdom (UK) stipulated that one month into lockdown orders, mental distress levels well exceeded the predicted trajectories of previous years [13].

As the COVID-19 pandemic is ongoing, the long-term mental health effects are not yet known [14]. During the SARS outbreak, a range of literature concluded that the mental health consequences of SARS were not entirely immediate and lagged in comparison to the infectious outbreak [9,15,16,17]. Psychological distress among SARS survivors displayed a 64% prevalence one year after the initial outbreak [9]. These results may be indicative of the effects we can expect from the current pandemic.

Studies exploring different population groups affected by COVID-19 have identified some common risk factors associated with a higher likelihood of developing anxiety symptoms, including: younger age groups, being female, having pre-existing mental health issues, and lower socioeconomic status (SES) populations [18,19]. The effects of COVID-19 on healthcare workers, the general population, and other vulnerable groups such as pregnant women have been well documented. Reviews conducted on the comparison between health care workers and the general population have been extensive. However, no review comparing multiple different groups, namely, that of healthcare workers, the general population, university students, and other vulnerable groups (pregnant women, the elderly, teachers, and police) currently exists.

The present study aims to, (1) systematically review and identify the prevalence and associations of anxiety in COVID-19 within multiple affected populations, and (2) identify common risk factors across the population groups, to aid in the treatment of global mental health. The identification of vulnerable groups may aid in developing stronger accuracy in intervention strategies for future pandemics.

## 2. Methods 

This qualitative systematic review was conducted to compare the anxiety levels amongst different sample populations in relation to the COVID-19 pandemic. The present review was structured on the Preferred Reporting Items for Systematic Reviews and Meta-analyses (PRISMA) criteria [20].

### 2.1. Eligibility Criteria 

The inclusion of only full peer-reviewed journal publications with available full text was sourced for the present review. Only papers published within the last two years (2020–2021) were included. The purpose of the implementation of this timeframe was to limit the results to the COVID-19 pandemic. Non-English language publications and papers with formats such as letters to the editor, books/book chapters, short commentaries, review articles, news releases, and research highlights were excluded.

Further exclusions included any studies on participants less than 18 years of age and those focused on populations containing comorbidities. Qualitative and mixed-method studies were also excluded from the study. Reasons pertaining to this exclusion include a higher likelihood of methodological bias and difficulties, as well as issues relating to the appraisal and synthesis of such data [21]. Studies that implemented self-made, unvalidated methods such as questionnaires were also excluded to ensure the papers included were of a uniform standard. The final mode of exclusion was based on sample size. The initial search on COVID-19 yielded many results to confirm that only the most vital papers were included; any studies with sample sizes less than 1000 were excluded [22].

### 2.2. Literature Search 

The review period commenced in April of 2021, and was followed by further updates in May, June, and July. The final search was updated on the 5 July 2021. Papers reporting the prevalence of anxiety in COVID-19 were selected for the review. The databases selected for the search were EMBASE, OVID MEDLINE, PsycINFO, and CINAHL. These databases were chosen as they are likely to yield the most relevant results targeting the research question and selection criteria. The relevance of these databases is attributed to their comprehensive coverage and inclusion of various academic journals. Table 1 illustrates the full search strategy implemented for each database.

### 2.3. Study Selection

The total number of search results from all four databases were imported into Endnote version 20.1 (Clarivate ^TM^, Sydney, Australia) software. A final number of 3537 journal articles were imported for the review on COVID-19 and anxiety. Figure 1 summarises the methodology and demonstrates the steps taken to derive the final number of papers. During the identification phase, 3537 papers were identified as relevant to the search terms, and a total of 547 duplicate papers were removed. Following the subsequent screening, 2990 studies were screened and from these, 2822 were excluded for various reasons. Reasons for exclusion included studies not meeting the inclusion criteria (44), being outside of the scope of the project (19) and not reporting on anxiety (18). One hundred and sixty-eight papers were sought for retrieval in full text, with a further 81 articles excluded for reasons detailed in Figure 1. A total of 87 papers were deemed eligible for inclusion in the present review.

### 2.4. Quality Assessment 

Two authors, H.S and A.E, screened the studies in full text to determine the eligibility for inclusion. Any dispute in the inclusion of studies were resolved as the authors came to an agreement. The study design, quality, and methods were compared against The Joanna Briggs Institute (JBI) critical appraisal tool to ensure an adequate standard to be included in the review [23]. The JBI critical appraisal tool provided varying checklists depending on the nature and design of the paper, with the most utilised checklist in this review being the checklist for analytical cross-sectional studies, as the majority of the selected papers used a cross-sectional study design.

### 2.5. Data Extraction and Synthesis 

The studies were imported into Microsoft Excel version 16.54 (Microsoft, Sydney, Australia), where the data for the results were extracted. The sample sizes, other study characteristics, study design, psychometric scores, results, and main findings of each study were extracted for the review. The results were collated into groups corresponding to the different population types.

## 3. Results 

### 3.1. Study Characteristics 

The sample size assessed in this review, derived from the total sample size of each study included in the review, was *n* = 755,180 with approximately *n* = 432,944 females, *n* = 280,089 males, and *n* = 42,147 participants that identified as other or did not report their sex. The age range of individuals within the included papers was 18–100 years and encompassed participants from 32 countries, with the highest number of studies originating from China (26/87 studies). The majority of studies were cross-sectional in design (70 studies), followed by longitudinal studies (13 studies), cohort studies (3 studies), and one case-control. All studies utilised validated psychometric measures, with the most common measure being the generalised anxiety disorder (GAD) 7 item scale [24] (43 studies). Other psychometric measures utilised included the Depression Anxiety Stress Scale (DASS) [20], the Patient Health Questionnaire (PHQ) [25], and the State-Trait Anxiety Inventory (STAI) [26]. Key differences in these measures are the extent to which anxiety is assessed, with measures such as the GAD and STAI focusing on generalised anxiety disorder only and state-related anxiety, respectively. Measures such as the DASS and PHQ evaluate other mental health symptoms relating to stress and depression, and anxiety symptoms. A summary of the study characteristics and anxiety prevalence is detailed in Table 2.

### 3.2. The General Population Group

The general population was the most common group studied amongst the studies included in the review, with 47 papers focusing on anxiety assessment. The 47 papers comprised of a sample size of *n* = 421,598 participants, with *n* = 208,675 females, *n* = 178,187 males, and *n* = 34,736 other or sex not reported. The prevalence of anxiety ranged from 3.4–97.47% across the 47 study populations. The overall pooled anxiety prevalence was 34%, although eight studies did not directly report the prevalence of anxiety in their populations.

Amongst the general population, three studies [27,39,89] demonstrated that the prevalence of anxiety during the COVID-19 pandemic had risen when compared to data from preceding years; that is, in 2017 anxiety rates were 6%, but after the pandemic hit, this figure inflated to 19% [89]. Conversely, Velden (2020) reported no significant increase in the prevalence of anxiety in a before and after study comparing mental health rates in 2019 and 2020 [100]. However, the authors did note that despite an absence of an increase in anxiety, the risk factors predisposing participants to mental distress had changed since the onset of the pandemic, leaving students, job seekers, those with children, and those who housekeep more at risk in 2020 compared to the previous year.

Geographical locations that were identified as COVID-19 epicentres had higher instances of anxiety compared to non-epicentre areas [27,28,35,42,72,81,110]. Moreover, COVID-19 prevalent areas that exemplified elevated testing rates reported decreased anxiety [99]. Those with increased contact with COVID-19 infected individuals exhibited stronger associations with anxiety [45,87,94], especially if the individual was exposed to COVID-19 in a working environment such as healthcare [56,67]. Populations infected with COVID-19 expressed more anxiety than those who were not infected [44,56,61,102]. Job loss or financial hardship due to COVID-19 was often a predictor or factor for worse anxiety [39,93].

Quarantine and lockdown orders proved detrimental to mental health, as demonstrated in ten studies [27,37,42,45,82,85,86,99,101], with increased loneliness and isolation being the cause of significant increases in anxiety. In an Australian longitudinal study [32], there was a 23% increase in anxiety over a 12-week restriction period. Quarantining alone resulted in lower anxiety than people isolating with elderly dependents [35]. Three studies concluded that anxiety levels in populations decreased when rules were eased or when participants were exempted from participating in quarantines [11,35,75].

Certain demographic groups were identified as having a higher prevalence of anxiety or being more at risk of developing adverse mental health issues. Twenty-two studies found that females consistently had higher levels of anxiety than males [11,28,31,32,33,35,39,41,62,63,65,75,79,82,86,88,92,94,96,102,105,110]. However, two studies found that males were more anxious when living with dependents under 18 [50,61] and that younger males had higher instances of anxiety [56]. One study reported that males had higher rates of anxiety than females overall [107]. Two studies [82] and [101] did not delineate any significant differences between the sexes. Five studies reported that lower socioeconomic status was representative of greater anxiety [37,45,67,70,101]. Prior mental illness was also a contributing factor for worse mental health after COVID-19 [39,44,63,65,97]. Younger age groups displayed more anxiety than older age groups in sixteen studies [28,32,37,39,42,45,61,62,85,86,89,94,97,101,102].

Contrastingly, four studies identified an opposite trend, with elderly and older populations experiencing more anxiety than younger groups [46,79,82,90]. Six studies identified having a higher education being associated with worse anxiety [33,37,47,66,67,101], while two studies identified that lower education equated to increased anxiety [86,97]. Living alone or remotely and being unemployed were influences on increased anxiety [45,65,89,97]. Conversely, Fu and colleagues (2020) indicated that living in a city may be predictive of worse mental health [46]. Two studies reported no difference in anxiety levels between different demographics, including sex, age, education, or socioeconomic status [87,92].

### 3.3. Healthcare Worker Group 

Healthcare workers constituted the subject of 25 of the 87 studies included in this review, with a total sample size of 43,387 participants. This sample consisted of *n* = 32,185 females, *n* = 9675 males, and *n* = 1527 participants who identified as other. The prevalence of anxiety ranged from 13.3%–100% in all study populations, with a pooled prevalence of 36%.

Five studies found that the prevalence of anxiety was higher in healthcare workers than in other professions, and this included clinical, non-clinical, and administrative healthcare workers [30,40,78,80,111]. A greater prevalence of anxiety was found in frontline healthcare responders compared to second-line or non-COVID-19 healthcare workers, and this was highlighted in twelve papers [29,30,34,43,48,51,53,69,74,76,109,113]. This was further endorsed, as healthcare staff not working in COVID-19 epicentres scored lower for anxiety [57]. Amongst clinical healthcare workers, more studies found that nurses suffered to a greater level from anxiety than physicians [53,69,73,95]. However, this was countered by Lie and colleagues [74], where it was found that physicians displayed more anxiety-like symptomology than nurses. Non-clinical healthcare workers, such as administrative staff and clerks, scored higher on anxiety psychometric measurements than clinical staff [38,51,58]. One study contradicted this, suggesting that anxiety in clinical staff was more significant than that that observed in non-clinical staff [76].

A lack of resources, including testing equipment and personal protective equipment (PPE), increased the likelihood of anxiety symptoms amongst hospital staff [104,113]. Additional anxiety was promoted by the worry of infecting family members with COVID-19 or being infected themselves [69,77], hence there was a strong association between job risk and anxiety [95]. Hacimusalar and colleagues found that situational anxiety was much higher in healthcare staff, whereas general anxiety was more common in the broader population [53]. During subsequent waves of COVID-19 infection, anxiety levels worsened among healthcare workers [52]. The increased demand in working hours exposed healthcare workers, both clinical and non-clinical, to be more at risk [74,113]. The occurrence of medical violence during peak COVID-19 periods also exacerbated mental health conditions. In ten studies, females were found to have increased levels of anxiety [38,48,51,57,69,77,95,104,109,113]. Five papers reported that younger healthcare workers such as trainees experienced more anxiety than older workers [48,51,69,73,113], but others reported that older healthcare workers were the more affected group [57,58,109]. The existence of a prior mental health illness or living alone were also reported as significant risk factors [58,73,104].

### 3.4. University Students 

Eight papers focused on the prevalence of anxiety in university students [36,47,54,60,68,83,98,106]. The total sample size of the student group was *n* = 183,390, with *n* = 113,504 females, *n* = 64,114 males, and *n* = 2772 participants who identified as other. The prevalence of anxiety ranged from 0–71.5% in all study populations, with the pooled prevalence being 34.7%.

Islaml and colleagues (2020) reported that anxiety amongst university students had worsened compared to pre-pandemic rates and with the duration of lockdowns. Conversely, Kim et al., (2021) reported no significant changes in anxiety throughout lockdowns [68]. Two papers denoted adverse anxiety related to worry about academics and dissatisfaction with COVID-19 distance learning measures [36,60]. The impact of restrictions on daily life was proven detrimental to anxiety symptoms [36,83]. The implications of lockdowns resulted in increased loneliness and lack of social support, and both of these factors were uncovered to be responsible for a rapid increase in clinical anxiety scores [36,46]. Although restrictive orders caused some populations to experience more anxiety, another study showed that self-efficacy as a result of isolation decreased anxiety [98]. Living in a COVID-19 hotspot or personally knowing an infected person were predictors of higher anxiety [54,106]. Sun and colleagues (2021) found that the threat of being infected with COVID-19 and the stigma associated with that caused university students to be more anxious about contracting the infection [98]. Being exposed to more news and to COVID-19 related social media was strongly associated with worsened anxiety [98,106]. Financial instability caused by the pandemic was a significant factor for increased anxiety in four studies [36,47,98,106]. Further, residing with more than five family members was also predictive of anxiety [54]. Five studies identified female students as having higher scores of anxiety compared to male students [47,54,98,106]. Two studies found that postgraduate students aged in their mid-to-late 20s had higher levels of anxiety when compared to undergraduates [47,60]. This was opposed by Odriozola-Gonzalez and colleagues (2020), where it was established that undergraduate students were more anxious than postgraduates [83].

#### 3.4.1. Other Adults of the General Population

The remaining seven studies focused on multiple different groups, including parents, teachers, the elderly, police and pregnant women, in which the effects of COVID-19 on anxiety level varied, as detailed below.

##### Anxiety in Parents

Johnson and colleagues (2021) conducted a longitudinal study on the mechanisms of parental distress during the COVID-19 pandemic [64]. This study had a sample size of *n* = 2868, consisting of *n* = 2278 females and *n* = 590 males. They found that at T1, when lockdowns were strictest, 23.3% of participants met the clinical cut-off for generalised anxiety, and at T2, when restrictions were being eased, anxiety prevalence was lowered to 13.8% [64]. Anxiety was also higher in females than males (T1: 25.7% vs. 14%) [64].

##### Anxiety in Teachers 

Two studies focused on teachers with a combined sample size of *n* = 90,244, with *n* = 69,462 females and *n* = 20,772 males. The pooled prevalence of anxiety in both populations was 27.2%, with either 49.5% [84] or 26.6% [71] of participants reporting COVID-19 related anxiety. In both studies, female teachers experienced more anxiety than male teachers and older teachers more so than younger teachers.

##### Anxiety in the Elderly Population 

Two studies focused on the elderly with a sample size of *n* = 8766, with *n* = 4817 females and *n* = 3791 males [49,91]. Both studies concluded that those living alone, experiencing financial hardship, not exercising, and being widowed indicated increased anxiety. Robb and colleagues (2020) reported that with every five-year increase in age group within the study population, there was a 22% decrease in anxiety results [91]. This was contrasted in a study by Garcia-Fernandez and colleagues (2020) [49], which found no differences in anxiety based on age. Thirty four percent of participants reported anxiety when they scored within the normal clinical range [91].

##### Anxiety in Police 

Yuan and colleagues (2020) investigated the psychological impact of COVID-19 on police officers in a sample size of *n* = 3517, with *n* = 557 females and *n* = 2960 males [108]. Of this population group, 8.79% reported moderate to severe anxiety, with older, more educated officers residing in or near a city having higher anxiety levels [108]. Males had a lower frequency of anxiety than females (34.1% vs. 37.7%) [108].

##### Anxiety in Pregnant Women

Zilver and colleagues (2021) assessed a sample of *n* = 1466 pregnant women and found a 19.5% prevalence of anxiety in the study sample, but the study concluded this was not a significant increase compared to anxiety rates before the pandemic [112]. Table 3 summaries the results comparing the main findings of the review.

## 4. Discussion 

There have been many recent systematic reviews published on the mental health effects of the COVID-19 pandemic. The majority of these studies however, focus on specific sample populations [114]. Wu and colleagues (2021) completed a systematic review of various mental health outcomes related to COVD-19 in multiple sample groups [14]. However, this review was limited to the early phase of the pandemic (January–March, 2020) and mostly was contained to China [14].

The results of this systematic review demonstrate that the COVID-19 pandemic has negatively impacted the mental health of many populations in society. Anxiety is prevalent within the general population, healthcare workers, university students, and other vulnerable groups [28,40,51,60,62], and the onset of COVID-19 has exacerbated it [90]. The main contributors to this observed increase in anxiety are unique to this current outbreak alone. The implementation of stringent global lockdowns and quarantine orders have been one of the primary methods to achieve infection control. Although proven as effective measures to reduce transmission and COVID-19 case numbers, they have brought about great mental turmoil globally [59].

Social isolation was common during previous episodes of infectious disease outbreaks such as the quarantining of populations during the SARS and Ebola outbreaks, although this was mostly restricted to those infected or in contact with the disease [115,116]. However, the COVID-19 pandemic has set a new precedent in this regard as orders of social isolation, quarantine, and lockdowns have, to some level, been imposed upon the majority of the world’s populations. The literature indicates that individuals with otherwise good mental health at the start of lockdown experienced mental decline the longer and more stringent the lockdown was [117]. This coincides with the findings of this systematic review, which demonstrates that quarantine and lockdown orders increased the instances of loneliness and isolation, which in turn promoted anxiety levels. Sharma and colleagues (2020) found that 50% of participants displayed anxiety symptoms after being subjected to quarantine [118]. This alarmingly high figure is indicative of a more significant issue at hand, demonstrating that the support networks in place are lacking. As apparent in the recent, more than 100-day (June–October) lockdown in Sydney in 2021, the mental health risk associated with longer more stringent lockdowns could see anxiety cases reach a much higher level, should such lockdowns continue into the future.

Alternatively, some studies indicate that lockdown and quarantine orders have a small or no impact on mental health [100]. However, these findings can be explained by the limited sample size in some of these studies, which did not include a wide range of socioeconomic diversity and a had a degree of heterogeneity in the data [119].

### 4.1. Anxiety before and after COVID-19

The majority of papers in the present systematic review found that the prevalence of anxiety was higher in 2020 when compared to the rates of previous years (2019) [34,37,99]. The Australian Institute of Health and Welfare (AIHW) reported that COVID-19 related restriction on movement, physical and social isolation, loss of employment, and other adverse effects of the lockdowns resulted in an 18.4% and 30.7% increase in calls to Lifeline and Beyond Blue, respectively [120]. The call volume had increased compared to the volume of calls received at the same time the previous year in 2019 [120]. Following the onset of the COVID-19 pandemic, the Australian Government implemented a range of mental health services under the Medicare Benefits Schedule (MBS), which included subsidising telehealth services [120]. The AIHW reported that after the new telehealth items were added to the MBS, there was a high uptake in the number of people accessing these services [120].

### 4.2. Anxiety in Different Populations during COVID-19

The results indicate that COVID-19 affected anxiety levels in all of the different study populations evaluated (general population, healthcare workers, university students, teachers, pregnant women, the elderly, parents, and police). The degree of anxiety varied, as groups such as healthcare workers, females, and younger populations were more vulnerable than others [121,122]. During COVID-19, the overall prevalence of anxiety was highest in the initial stage of the outbreak, with the highest rate among healthcare workers (36%), followed by university students (34.7%), and the general population (34%). Among the other groups, teachers experienced the most anxiety (27.2%), compared to police officers, who had the lowest prevalence (8.79%). As discussed below, many factors are attributed to the variation in anxiety levels among different study samples.

#### 4.2.1. Anxiety in the General Population 

In this systematic review, the prevalence of anxiety among the general population (34%) coincided with the prevalence of anxiety found in other studies [65]. A systematic review concluded that the prevalence of anxiety in 103 studies of the general population was 27.3% [65]. Other studies reported levels as low as 21.6% [123] or as high as 81.9% [124].

The present study found that anxiety was significantly higher in populations living in epicentre regions, such as Wuhan, China [109]. This is supported by Zhao and colleagues (2020) [125], who found that those who resided within high infection areas, such as Hubei, China, displayed higher moderate to severe anxiety rates than those who lived in lower epidemic areas (less affected regions of mainland China) [125]. The increased health-related anxiety can explain this phenomenon in regions of more significant infectious outbreaks [125]. The escalation of health anxiety was predictive of generalised anxiety during the COVID-19 pandemic [126]. Within epicentre regions, additional testing carried out above the average rate resulted in a marked reduction in population anxiety [127], reducing the overall health anxiety and exemplifying a control over the outbreak. Increased exposure to COVID-19 was an indicator of worse anxiety, whether through casual contacts, workplace environments, or being infected with COVID-19 directly [128]. Literature suggests that exposure to COVID-19 infection results in a much higher prevalence of anxiety, especially if the contact is through family members [59]. Huang and colleagues (2020) reported that of the populations presenting with COVID-19 related anxiety, those with higher contact histories and those with confirmed infections displayed an elevated risk of anxiety symptoms [59].

Sex was a major determinant for anxiety amongst the general population, with twenty-two studies finding that females experienced significantly higher anxiety levels than males. Multiple studies support these findings, suggesting that females do, in fact, experience higher levels of mental distress and anxiety concerning COVID-19 [65,121,129]. Evidence demonstrates that this increased effect on females could be attributed to the burden many females feel as primary caregivers. With the added stressor of the pandemic, females are more likely than males to care for dependent family members [130]. Fu and colleagues (2020) also suggested that females were more likely to score positive for anxiety because they were more likely to convey their emotions than males [46]. Divergencies in neurochemistry may expose females to a slightly heightened risk of developing anxiety disorders [46]. One study analysed in this review found that males had experienced higher levels of anxiety than females [107]. This can be attributed to the decreased likelihood of males to seek mental health assistance due to perceived stigma [131]. An additional two studies found that although females experienced higher anxiety levels overall, males who care for dependents under the age of 18 had higher instances of anxiety than other male groups [50,61]. The additional stress of caring for young children during lockdown whilst working from home can explain this trend [64].

Socioeconomic status was another contributor to the severity of anxiety, with the COVID-19 related lockdowns resulting in a peak unemployment rate of 7.5%, the highest rate in the last 20 years, as reported by the ABS [12]. A multitude of studies found that job loss as a result of COVID-19 was a major contributor to significant surges in anxiety and attributed financial instability as a leading cause of a myriad of other severe mental health issues [132,133]. The present review also found that those with pre-existing mental health issues were at a heightened risk of aggravating their conditions. These findings are supported within the current literature, as the implication of quarantine and restriction has disrupted the routines of daily life many individuals rely on to uphold their mental health [134,135]. As access to health services has been restricted due to the pandemic, there has been a marked escalation in relapses of anxiety attacks and disorders [136].

Age was yet another factor linked to heightened anxiety levels, with the majority of included papers identifying younger age groups as more at risk for anxiety [11,35,39,45,54,68]. Recent findings have also concluded that younger age groups have higher rates of anxiety, as they often experience more financial and employment instability than older groups [137]. In conjunction with this, younger age groups are much more likely to consume more media coverage of the pandemic than older groups, with up to 3 h of social media exposure a day. This increased exposure has been found to increase anxiety odds by up to 3 times [54,137]. However, four studies identified higher anxiety levels in older groups [47,79,82,90], which can be explained by older groups being more likely to suffer from more extreme effects of COVID-19 [138]. The vulnerability of older populations is evident as mortality rates of those aged over 70 are upwards of 22.8% compared to a rate of 1.1% for those aged below 50 (Bonanad et al., 2020). This increased mortality rate is directly linked to worse psychological outcomes, with increased occurrences of death anxiety (Khademi et al., 2021).

#### 4.2.2. Anxiety in Healthcare Workers 

The prevalence of anxiety experienced by healthcare workers was the highest rate amongst all the population groups, with a pooled prevalence of 36% from 25 studies. This finding is greater than the frequency found in the current literature. The prevalence in a systematic review on healthcare workers found that 23.2% of the population experienced anxiety [139]. An Indonesian study found a more similar prevalence of 33% [140].

Frontline healthcare workers were found to experience more anxiety than non-frontline healthcare workers and non-clinical healthcare staff (administrative clerks). This finding can be justified as studies illustrate that increased exposure to COVID-19 infection via a workplace setting is responsible for higher anxiety [30]. As frontline healthcare workers are at a greater risk of becoming infected, job anxiety is more prevalent in these populations than healthcare workers who have limited contact with infected patients (Cai et al., 2020). Due to the influx of hospitalisations related to COVID, healthcare staff have had to work longer hours with limited resources increasing their vulnerability to burnout and stress [53]. This has, in turn, drastically affected mental health, with reports of heightened anxiety found in frontline healthcare staff across many countries [18,141]. Non-frontline workers also had an increase in anxiety. However, frontline workers were more impacted, as the lack of hospital resources and diminished staffing due to need in COVID-19 wards caused a stretch in healthcare systems [140]. Some studies in the present review found that non-clinical healthcare workers presented with higher anxiety levels than clinical staff [38,51,58]; this was attributed to limited training in regard to infectious disease and crisis management [58]. It was found that upon completion of crisis training, the anxiety psychometric measures of non-clinical healthcare workers decreased drastically [58].

The fear of healthcare workers infecting their families was a major determinant for health and job-related anxiety. This is supported by Dai and colleagues (2020), who found that one of the greatest fears healthcare workers expressed was infecting others outside of the workplace [142]. Younger healthcare workers also expressed higher scores of anxiety, which could be explained by their lack of training and experience in the role [48,51,69,73,113]. This also coincides with findings of the general population, as younger age groups were found to be more at risk. However, three papers reported higher anxiety levels in older groups, with the vulnerability of older-aged populations to COVID-19 infection; the increased likelihood of older participants having dependants could explain this finding [57,58,109]. Similar to the results of the general population, females experienced higher anxiety than males amongst the healthcare workers.

#### 4.2.3. Anxiety in University Students 

The prevalence of anxiety among university students was 34.7%, which was close to the prevalence found in the general population (34%) and in line with the literature, as Halperin and colleagues (2021) reported anxiety prevalence among university students to be 30.6% [54].

Two studies conflicted in their findings on the prevalence of anxiety in university students before and after the pandemic [60,68]. The study that did not identify an increase in anxiety from before the pandemic highlights that introducing university aid and classes moving to pass/fail systems may have dampened the mental effects of COVID-19 [68]. Literature also suggests that the introduction of lockdowns has allowed students to focus on hobbies and get more sleep, as classes moved online [143]. In contrast to this, a plethora of studies have supported the finding that anxiety has increased significantly since the onset of the COVID-19 pandemic [47,54,60]. Students living on campus were found to have more anxiety symptoms than those who did not. The financial instability of living on campus while not being able to work to support themselves has caused many university students to become vulnerable to mental deterioration [54]. Literature also supports the finding that the increased loneliness experienced by students living on campus is determinative of higher anxiety psychometric scores [144].

Academic anxiety was a significant source of stress among university students. With the transition of classes to an online setting, the cracks in many education systems have begun to show [98]. The transition to online schooling has caused distress in many students who have issues with self-learning, which has caused an upsurge in anxiety related to academics and isolation, and a lowered perception of academic self-efficacy [145]. Due to the younger age demographic of university students, they consume more social media, akin to the younger age groups in the general population, and the mass consumption of COVID-19 related media indicates increased anxiety [11,121]. Parallel to the other population groups, those living in hotspot areas and females had higher levels of anxiety. The literature supports that female students were more likely to score positively for anxiety than male students [83,98]. Although females may experience higher anxiety for many reasons, the greater percentage of females that participate in studies may explain this phenomenon [146].

#### 4.2.4. Anxiety in Other Adults of the General Population 

There were seven papers assessing the other adult populations that varied in the severity of anxiety present [49,64,71,84,91,108,112]. The levels of anxiety found in the different sample populations had a direct correlation to the degree of vulnerability they experienced as a result of the COVID-19 pandemic.

Teachers had the highest prevalence of anxiety, with 49.5% of teachers reporting COVID-19 related anxiety [84]. The additional strain placed on education systems due to the closing of schools and online learning has resulted in teachers experiencing high levels of mental distress [147]. Contrastingly, the delayed closure of schools caused teachers to have increased anxiety regarding their safety and risk of contracting COVID-19 [148]. Parents had the second highest prevalence of anxiety, with 23.3% having anxiety induced by lockdowns [64]. Similar to teachers, the closure of schools exacerbated anxiety in parents as they were left responsible for their children’s education [64]. Due to lockdowns, movement outside of the home was limited to a necessity basis, such as grocery shopping or work; anxiety in parents was elevated due to the confinement of children within the home [64].

The elderly population did not have significant levels of anxiety and anxiety symptoms were found to be lowered by 22% as age increased [49,64,91]. The already limited mobility of older populations outside of the home promoted lower levels of anxiety as many did not perceive themselves to be at risk of transmission [50]. Anxiety was present in 19.5% of pregnant women, although this was not significant from pre-pandemic rates [112]. This was attributed to COVID-19 hospital interventions that allowed pregnant women to have their partners present while giving birth [112]. Finally, police officers were the least impacted group, exhibiting low anxiety rates at 8.79% [108]. The COVID-19 pandemic did not have an impact on police officers due to the overall compliance of the general population in adhering to regulations and lockdowns [108].

### 4.3. Limitations 

The strengths of the present review were in the extensive comparison of anxiety in multiple sample population groups. To the best of the authors’ knowledge, the comparison between the general population, healthcare workers, university students, teachers, parents, the elderly, pregnant women, and police officers has not been drawn before.

Although the present systematic review presents some important findings, various limitations were noted during the process. Firstly, restricting the review to only English language publications may have potentially introduced language bias into the study. Language bias is the phenomenon where studies of languages other than English, the predominant language utilised within research, may be overlooked and thus potentially limit the scope of the review [149]. Secondly, the sample size constraint implemented also posed a limitation. The exclusion of studies that did not meet the 1000 sample size criteria may have possibly excluded many relevant studies. As the COVID-19 pandemic is ongoing, the mental health effects are not fully characterised and are transforming as more literature is being published. In light of this information, this review was restricted to papers published before August 2021.

## 5. Conclusions

The COVID-19 pandemic has been found to have significantly contributed to worse anxiety in all populations studied. Those most exposed to infection, such as healthcare workers, are at risk of succumbing to immense mental pressure. If this is not remedied, a multitude of issues will arise, as a healthy state of mind is vital to the success of society [150]. Without addressing the high rates of anxiety, we may see the breakdown of healthcare systems struggling to cope, a general population havocked by economic and personal strain, and university students, the professionals of the future, being inflicted with mental anguish. Further longitudinal study is required to better understanding the factors and associations contributing to anxiety during pandemics, and will help guide such future outbreaks as well as prepare for emergency situations; this is critical for success in the future.

## Figures and Tables

**Figure 1 ijerph-19-02189-f001:**
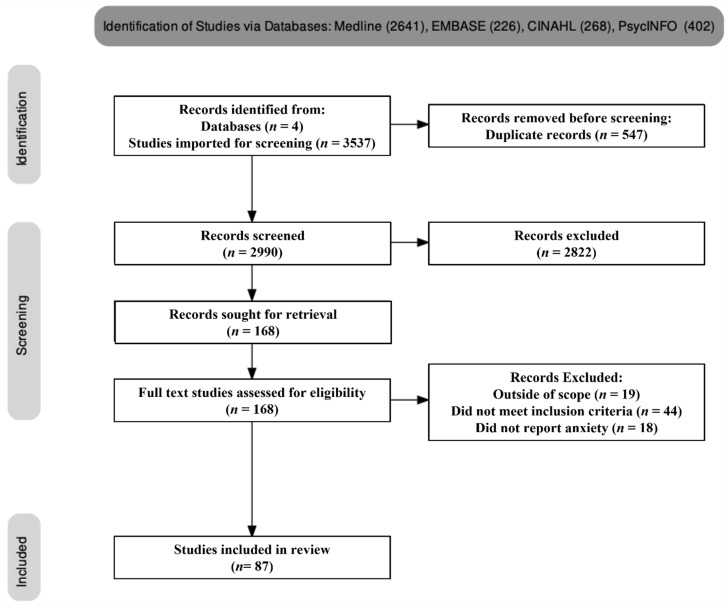
PRISMA flow diagram. The final number of papers included in the review was 87.

**Table 1 ijerph-19-02189-t001:** Search strategy implemented and results generated from each of the four databases utilised.

Database	Search Terms	Search Limiters	Result
EMBASE (Ovid)	(Coronavirus OR COVID-19) AND (Anxiety)	Journal Article English 2020–2021 No Medline Results	226
Medline (Ovid)	(Coronavirus OR COVID-19) AND (Anxiety)	Journal Article English 2020–2021	2641
CINAHL (EBSCO)	(Coronavirus OR COVID-19) AND (Anxiety)	Journal Article English 2020–2021 No Expanders	268
PsycINFO (EBSCO)	(Coronavirus OR COVID-19) AND (Anxiety)	Journal Article English 2020–2021 No Expanders	402

**Table 2 ijerph-19-02189-t002:** Characteristics and anxiety prevalence of the selected studies.

Reference	Study Design	Population Type	Country	Sample Size	Assessment Tools	Prevalence of Anxiety (%)
Aharon et al., 2020 [27]	Cross-sectional	General population	Israel and Italy	1015	PHQ-4, SF-8	50.2% of Italian and 42.2% of Israelis
Albagmi et al., 2012 [28]	Cross-sectional	General population	Saudi Arabia	3017	GAD-7	80% (mild), 11.4% (moderate), 8.2% (severe)
Alshekaili et al., 2020 [29]	Cross-sectional	Healthcare workers	Oman	1139	DASS-21	34.1%
Antonijevic et al., 2020 [30]	Cross-sectional	Healthcare workers	Serbia	1678	GAD-7	43.31% (minimal), 30.9% (mild), 12.99% (moderate),12.8% (severe).
Ausin et al., 2020 [31]	longitudinal	General population	Spain	1041	GAD-2	N/A
Batterham et al., 2021 [32]	longitudinal	General population	Australia	1296	GAD-7, PHQ-9	77%
Bendau et al., 2020 [11]	Longitudinal	General population	Germany	2376	GAD-2, PHQ-4	N/A
Budimir et al., 2021 [33]	Cross-sectional	General population	Austria and UK	2011	GAD-7	18.9% UK and 6% Austria
Cai et al., 2020 [34]	case-control	Healthcare workers	China	2346	BAI	Frontline 15.7%, non-frontline 7.4%
Canet-Juric et al., 2020 [35]	longitudinal	General population	Argentina	6057	STAI	N/A
Cao et al., 2020 [36]	Cluster Sampling	University Students	China	7143	GAD-7	Mild (21.3%), moderate (2.7%), severe (0.9%)
Chen et al., 2021 [37]	Cross-sectional	General population (quarantined)	China	1837	STAI	16.3%
Chew et al., 2020 [38]	Cross-sectional	Healthcare workers	India, Indonesia, Singapore, Malaysia and Vietnam	1146	DASS-21	India (0.8%), Singapore (3.6%), Vietnam (6.7%), Indonesia (6.8%) and Malaysia (14.9%)
Dawel et al., 2020 [39]	longitudinal	General population	Australia	1296	GAD-7, PHQ-9, WHO-5	N/A
Denning et al., 2021 [40]	Cross-sectional	Healthcare workers	UK, Poland and Singapore	3537	HADS	20%
Di Blasi et al., 2021 [41]	longitudinal	General population	Italy	1129	DASS-21	N/A
Di Giuseppe et al., 2020 [42]	Cross-sectional	General population	Italy	5683	SCL-90	51.1%
Di Mattei et al., 2021 [43]	Baseline assessment	Healthcare workers	Italy	1055	DASS-21	69.4%
Fiorillo et al., 2020 [44]	longitudinal	General population	Italy	20,720	DASS-21, GHQ	Moderate (16.7%) and severe or extremely severe (17.6%)
Fisher et al., 2020 [45]	Cross-sectional	General Population	Australia	13,829	GAD-7, PHQ-9	21%
Fu et al., 2020 [46]	Cross-sectional	General population	China	1242	GAD-7, PHQ-9	27.6%
Fu et al., 2021 [47]	Cross-sectional	University students	China	89,588	GAD-7	41.1%
Gainer et al., 2021 [48]	Cross-sectional	Healthcare workers	US	1724	GAD-7, PHQ-9	36.5%
Garcia-Fernandez et al., 2020 [49]	Cross-sectional	Elderly population	Spain	1639	HARS	N/A
Garcia-Fernandez et al., 2020 [50]	Cross-sectional	General population	Spain	1635	HARS	N/A
Giardino et al., 2020 [51]	Cross-sectional	Healthcare workers	Argentina	1059	DASS-18	76.5%
Gundogmus et al., 2021 [52]	longitudinal	Healthcare Workers	Turkey	2460	DASS-21	29.6%
Hacimusalar et al., 2020 [53]	Cross-sectional	Healthcare, non-healthcare	Turkey	2156	STAI	89.5%
Halperin et al., 2021 [54]	Cross-sectional	University students	US	1428	GAD-7, PHQ-9	30.6%
Hammarberg et al., 2020 [55]	Cross-sectional	General population	Australia	13,762	GAD-7	21.8% females, 14.2% males
Hassannia et al., 2021 [56]	Cross-sectional	Healthcare workers and general population	Iran	2045	HADS	65.6%
He et al., 2021 [57]	Cross-sectional	Healthcare workers	China	1971	GAD-7	29.3%
Hennein et al., 2021 [58]	Cross-sectional	Healthcare workers	US	1092	GAD-7	15.6%
Huang et al., 2021 [59]	Cross-sectional	Healthcare workers	Singapore	1638	GAD-7	12.5%
Islaml et al., 2020 [60]	Cross-sectional	University students	Bangladesh	3122	DASS-21	Mild anxiety (71.5%), moderate (63.6%), severe (40.3%) and very severe (27.5%).
Jacques-Avino et al., 2020 [61]	Cross-sectional	General population	Spain	7053	GAD-7	31.2% females, 17.7% males
Jia et al., 2020 [62]	Cross-sectional	General population	UK	3097	GAD-7	57% (26% moderate to severe anxiety)
Jiang et al., 2020 [63]	Cross-sectional	General population	China	60,199	SAI	Mild (33.21%), moderate (41.27%) and severe (22.99%).
Johnson et al., 2021 [64]	longitudinal	Parents	Norway	2868	GAD-7	N/A
Kantor and Kantor, 2020 [65]	Cross-sectional	General population	US	1005	GAD-7	52.1% mild, 26.8% anxiety disorder
Karaivazoglou et al., 2021 [66]	Cross-sectional	General population	Greece	1443	HADS	20%
Khubchandani et al. 2021 [67]	Cross-sectional	General population	US	1978	GAD-2, PHQ-4	42%
Kim et al., 2021 [68]	longitudinal	University Students	US	8613	GAD	No significant changes were found in the rates of anxiety from before the pandemic.
Lai et al., 2020 [69]	Cross-sectional	Healthcare workers	China	1257	GAD-7	44.6%
Lei et al., 2020 [70]	Cross-sectional	General population	China	1593	SAS	8.3%
Li et al., 2020 [71]	Cross-sectional	Teachers	China	88,611	GAD-7	13.67%
Li et al., 2021 [72]	Cross-sectional	General population	China	1201	DASS-21	34.2%
Liu et al., 2021 [73]	Cross-sectional	Healthcare workers	China	1090	GAD-7	13.3%
Liu et al., 2020 [74]	Cross-sectional	Healthcare workers (paediatric)	China	2031	DASS-21	18.3%
Lu et al., 2020a [75]	Cross-sectional	General population and frontline workers	China	1417	GAD-7	52.1% of the general public and 56% of frontline workers
Lu et al., 2020b [76]	Cross-sectional	Healthcare workers	China	2299	HAMA	22.6% of medical staff showed mild to moderate anxiety and 2.9% were severe
Luceno-Moreno et al., 2020 [77]	Cross-sectional	Healthcare workers	Spain	1422	HADS	58.6% healthcare workers presented with an anxiety disorder.
Mattila et al., 2020 [78]	Cross-sectional	Healthcare workers	Finland	1995	GAD-7	30% mild anxiety, 10% moderate and 5% severe anxiety.
Meesala et al., 2021 [79]	Cross-sectional	General population	India	1346	CAS-7	N/A
Mosheva et al., 2020 [80]	Cross-sectional	Healthcare workers	Israel	1106	PROMIS	52.8%
Ngoc Cong Duong et al., 2020 [81]	Cross-sectional	General population	Vietnam	1385	DASS-21	14.1%
Nkire et al., 2021 [82]	Cross-sectional	General population	Canada	6041	GAD-7	46.7%
Odriozola-Gonzalez et al., 2020 [83]	Cross-sectional	University students and workers.	Spain	2530	DASS-21, IES	21.34%
Ozamiz-Etxebarria et al., 2021 [84]	Cross-sectional	Teachers	Spain	1633	DASS-21	49.5% (8.1% extreme severe and 7.6% severe)
Ozamiz-Etxebarria et al., 2020 [85]	longitudinal	General population	Spain	1933	DASS-21	26.9%
Pandey et al., 2020 [86]	Cross-sectional	General population	India	1395	DASS-21	Anxiety prevalence was 22.4% in the second week and 26.6% in the third week of lockdowns
Passavanti et al., 2021 [87]	Cross-sectional	General population	Australia, Iran, China, Ecuador, Italy, Norway and the US	1612	DASS-21	44.7% (5.2% mild, 17.4% moderate, 5.8% severe and 16.3% extremely severe).
Pieh et al., 2021 [88]	Cross-sectional	General population	UK	1006	GAD-7	39%
Peih et al., 2020 [89]	Cross-sectional	General population	Austria	1005	GAD-7	19%
Planchuelo-Gomez et al., 2020 [90]	longitudinal	General population	Spain	4724	DASS-21	49.66%
Robb et al., 2020 [91]	Cross-sectional	Elderly population	UK	7127	HADS	N/A
Rossi et al., 2020 [92]	Cross-sectional	Healthcare workers and general population	Italy	24,050	GAD-7	21.25% in the general population, 18.05% in second line healthcare workers and 20.55% in frontline workers.
Ruengorn et al., 2020 [93]	Cross-sectional	General population	Thailand	2303	GAD-7	56.9%
Serafim et al., 2021 [94]	Cross-sectional	General population	Brazil	3000	DASS-21	39.7%
Shen et al., 2020 [95]	Cross-sectional	Healthcare Workers	China	1637	SAS	10.02%
Sinawi et al., 2021 [96]	Cross-sectional	General Population	Oman	1538	GAD-7	22%
Solomou et al., 2020 [97]	Cohort study	General population	Cyprus	1642	GAD-7	41% mild, 23.1% moderate-severe
Sun et al., 2021 [98]	Cross-sectional	University Students	China	1912	GAD-7	34.73%
Tang et al., 2020 [99]	Cross-sectional	General population	China	1389	GAD-7	70.78%
Van der Velden et al., 2020 [100]	Longitudinal	General population	Holland	3983	GAD-7	No significant anxiety found
Wang et al., 2021a [101]	Case-control	General population	China	1674	ADS	27% in quarantined, 11.2% in general population
Wang et al., 2021b [102]	Cross-sectional	Healthcare workers	China	1063	GAD-7	48.7% in patients, 25.7% general population, 13.3% healthcare
Wang et al., 2020 [103]	Cross-sectional	General, covid and health	China	49,015	DASS-21	10.02%
Wanigasooriya et al., 2021 [104]	Cross-sectional	Healthcare workers	UK	2638	PHQ-4	34.31%
Warren et al., 2021 [105]	Cross-sectional	General population	United States	5023	PHQ-4	14.4%
Wathelet et al., 2020 [106]	Cross-sectional	University Students	France	69,054	STAI	27.47%
Wu et al., 2020 [107]	Cross-sectional	General population	China	24,789	STAI	51.6%
Yuan et al., 2020 [108]	Cross-sectional	Police	China	3517	HADS	8.79%
Zhang et al., 2020a [109]	Cross-sectional	Healthcare workers	China	2143	GAD-7	14.23%
Zhang et al., 2020b [110]	Cross-sectional	General population	China	123,768	GAD-7	3.4%
Zhou et al., 2020 [111]	Cross-sectional	Healthcare workers	China	1705	SAS	45.4%
Zilver et al., 2021 [112]	Cohort study	Pregnant women	Holland	1466	GAD-7	19.5%

Key: GAD-7, Generalised Anxiety Disorder—7 Item Scale; DASS-21, Depression Anxiety Stress Scale—21 Item; PHQ-4, Patient Health Questionnaire—4 Item; SAS, Self-Rating Anxiety Scale; HARS. Hamilton Anxiety Rating Scale; SCL-90, Symptom Checklist—90 Item; CAS, Coronavirus Anxiety Scale; PROMIS, Patient-Reported Outcomes Measurement Information System; STAI, State-Trait Anxiety Inventory; HADS, Hospital Anxiety and Depression Scale.

**Table 3 ijerph-19-02189-t003:** Summary and comparison of results.

Population Type	No. of Papers	Sample Size	Anxiety Prevalence	Main Findings
General Population	47	423,341	34%	Those in epicentres or those with higher exposure to COVID-19 are more at risk of developing anxiety. Quarantine had mass adverse effects on mental health with females, younger people, the elderly, and lower SES disproportionality impacted.
Healthcare Workers	25	43,387	36%	Increased working demands of COVID-19 have resulted in increased anxiety and burnout, especially in frontline workers. Health anxiety is highly prevalent with fears of infecting others. Females, trainee staff, and those with pre-existing conditions were most effected.
University Students	8	183,390	34.7%	Worry regarding academics resulted in a marked increase in anxiety, especially during periods of lockdowns and when compared to pre-pandemic times. Financial instability and stigma accessing aid may have contributed to this. Females were also identified as having higher anxiety than males.
Other Groups:	Elderly (2) Teachers (2) Parents (1) Pregnant (1) Police (1)	106,861	N/A	The other groups affected all exemplify groups within society that are vulnerable, with females experiencing more anxiety than males in all groups apart from the elderly population, where no difference was seen.

The number of papers pertaining to each population within ‘other groups’ are indicated in the brackets.

## Data Availability

No new data were created or analyzed in this study. Data sharing is not applicable to this article.

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
