# Peer review of "Anxiety Linked to COVID-19: A Systematic Review Comparing Anxiety Rates in Different Populations"

_ijerph, 2022, doi:10.3390/ijerph19042189_

Round 1
Reviewer 1 Report
Current study by Saeed et al is a systematic review dealing with studies related to anxiety linked to Covid-19. Authors have done literature search and have gathered several studies on anxiety and Covid19 done in various regions of the world. Current review has certain merits such as larger sample size in the studies which they have cited in the review. The review is exhaustive and great starting point for researchers who want to further design studies based on this literature review.
I have following minor comments-
- Please define anxiety (line 35) based on suitable criteria such as DSM-5 criteria and give reference for that.
- Studies cited in the review shows various psychometric measures such as GAD. It will be nice if authors can add few lines about other measures such as DASS, SCL etc to give a brief overview of differences between these psychometric measures.
Reviewer 2 Report
The current review is very well written and included very interesting and comprehensive information about the prevalence of anxiety in relation to COVID-19. According to the summary of the current review, healthcare workers had the highest prevalence of anxiety which was also dependent on the type of job in healthcare as nurse, doctor or administrative. Importantly, one of the important limitations which is including only English papers is mentioned in the text.
Author Response
Thank you for the feedback
Reviewer 3 Report
1. In the introduction part, it is desirable to add some physical illnesses related to anxiety and mental health, to complement the references concerning the COVID-19 disease.
2. Improve the information in Table 2 – Characteristics and Anxiety Prevalence of the Selected Studies, because there are many results and it is not easy and accessible for the reader to be able to compare them.
3. In the results part, a lot of information is presented, which is good, but graphs or tables could be made to visually make it easier to compare the results. For example in the paragraph:
Geographical locations that were identified as COVID-19 epicenters 238 had higher instances of anxiety compared to non-epicentre areas (Aharon 239 et al., 2021; Albagmi et al., 2021; Canet-Juric et al., 2020; Di Giuseppe et 240 al., 2020; Li et al., 2021; Ngoc Cong Duong et al., 2020; Zhang et al., 2020b). 241 Moreover, COVID-19 prevalent areas that exemplified elevated testing 242 rates reported decreased anxiety (Tang et al., 2020). Those with increased 243 contact with COVID-19 infected individuals exhibited stronger 244 associations with anxiety (Fisher et al., 2020; Passavanti et al., 2021; 245 Serafim et al., 2021), especially if the individual was exposed to COVID246 19 in a working environment such as healthcare (Hassannia et al., 2021; 247 Khubchandani et al., 2021). Populations infected with COVID-19 248 expressed more anxiety than those who were not infected (Fiorillo et al., 249 2020; Hassannia et al., 2021; Jacques-Avino et al., 2020; Wang et al., 250 2021b). Job loss or financial hardship due to COVID-19 was often a 251 predictor or factor for worse anxiety (Dawel et al., 2020; Ruengorn et al., 252 2021).
There is a lot of information that could be arranged in a different way.
